# A Priori Estimates of the Generalization Error for Two-layer Neural Networks

## Abstract

New estimates for the generalization error are established for a nonlinear regression problem using a two-layer neural network model. These new estimates are a priori in nature in the sense that the bounds depend only on some norms of the underlying functions to be fitted, not the parameters in the model. In contrast, most existing results for neural networks are a posteriori in nature in the sense that the bounds depend on some norms of the model parameters. The error rates are comparable to that of the Monte Carlo method in terms of the size of the dataset. Moreover, these bounds are equally effective in the over-parametrized regime when the network size is much larger than the size of the dataset.

## 1 Introduction

One of the most important theoretical challenges in machine learning comes from the fact that classical learning theory cannot explain the effectiveness of over-parametrized models in which the number of parameters is much larger than the size of the training set. This is especially the case for neural network models, which have achieved remarkable performance for a wide variety of problems (Bahdanau et al., 2015; Krizhevsky et al., 2012; Silver et al., 2016). Therefore, understanding the mechanism behind these successes is critical, which requires developing new analytical tools that can work effectively in the over-parametrized regime (Zhang et al., 2017).

Our work is partly motivated by the situation in classical approximation theory and finite element analysis. There are two kinds of error bounds in finite element analysis depending on whether the target solution (the ground truth) or the numerical solution enters into the bounds. Let $f^*$ and $\hat{f}_n$ be the true solution and the "numerical solution", respectively. In "a priori" error estimates, only norms of the true solution enter into the bounds, namely

$$\|\hat{f}_n - f^*\|_1 \leq C\|f^*\|_2.$$

In "a posteriori" error estimates, the norms of the numerical solution enter into the bounds:

$$\|\hat{f}_n - f^*\|_1 \leq C\|\hat{f}_n\|_3.$$

Here $\|\cdot\|_1, \|\cdot\|_2, \|\cdot\|_3$ denote various norms.

In this language, most recent theoretical efforts (Neyshabur et al., 2015; Bartlett et al., 2017; Golowich et al., 2018; Neyshabur et al., 2017; 2018a;b) on estimating the generalization error of neural networks should be viewed as "a posteriori" analysis, since all the bounds depend on some norms of the solutions. Unfortunately, as observed in Arora et al. (2018) and Neyshabur et al. (2018b), the numerical values of these norms are always huge, yielding vacuous estimates.

In this paper we pursue a different line of attack by providing "a priori" analysis. For this purpose, a suitably regularized two-layer network is considered. It is proved that the generalization error of regularized solutions is asymptotically sharp with constants depending only on the properties of the target function. Numerical experiments show that these a priori bounds are non-vacuous (Dziugaite & Roy, 2017) for datasets of practical interests, such as MNIST and CIFAR-10. In addition, our experimental results also suggest that such regularization terms are necessary in order for the model to be "well-posed" (see Section 6 for the precise meaning).

## 1.1 SETUP

We will focus on the regression problem. Let $f^* : \Omega \mapsto \mathbb{R}$ be the target function, with $\Omega = [-1, 1]^d$, and $S = \{(\mathbf{x}_i, y_i)\}_{i=1}^n$ denotes the training set. Here $\{\mathbf{x}_i\}_{i=1}^n$ are i.i.d samples drawn from an underlying distribution $\pi$ with $\text{supp}(\pi) \subset \Omega$, and $y_i = f^*(\mathbf{x}_i) + \varepsilon_i$, with $\varepsilon_i$ being the noise. Our aim is to recover $f^*$ by fitting $S$ using a two-layer fully connected neural network with ReLU (rectified linear units) activation:

$$f(x; \theta) = \sum_{k=1}^m a_k \sigma(\boldsymbol{b}_k \cdot \mathbf{x} + c_k), \tag{1}$$

where $\sigma(t) = \max(0, t)$ is the ReLU function, $\boldsymbol{b}_k \in \mathbb{R}^d$, and $\theta = \{(a_k, \boldsymbol{b}_k, c_k)\}_{k=1}^m$ represents all the parameters to be learned from the training data. $m$ denotes the *network width*. To control the complexity of networks, we use the following scale-invariant norm.

**Definition 1** (Path norm). For a two-layer ReLU network (1), the path norm is defined as

$$\|\theta\|_{\mathcal{P}} = \sum_{k=1}^m |a_k|(\|\boldsymbol{b}_k\|_1 + |c_k|).$$

**Definition 2** (Spectral norm). Given $f \in L^2(\Omega)$, denote $F \in L^2(\mathbb{R}^d)$ as an extension of $f$ to $\mathbb{R}^d$. Let $\hat{F}$ denote the Fourier transform of $F$, then $f(\mathbf{x}) = \int_{\mathbb{R}^d} e^{i\langle \mathbf{x}, \boldsymbol{\omega} \rangle} \hat{F}(\boldsymbol{\omega}) d\boldsymbol{\omega} \; \forall \mathbf{x} \in \Omega$. We define the spectral norm of $f$ as follows

$$\gamma(f) = \inf_{F \in L^2(\mathbb{R}^d), F|_\Omega = f|_\Omega} \int_{\mathbb{R}^d} \|\boldsymbol{\omega}\|_1^2 |\hat{F}(\boldsymbol{\omega})| \, d\boldsymbol{\omega}. \tag{2}$$

**Assumption 1** (Target function). *Following* Breiman *(1993) and* Klusowski & Barron *(2016), we consider target functions that have finite spectral norm. By defining*

$$\mathcal{F}_s := L^2(\Omega) \cap \big\{ \, f(\mathbf{x}) : \Omega \mapsto \mathbb{R} \,\big|\, \gamma(f) < \infty, \, \|f\|_\infty \leq 1 \big\}, \tag{3}$$

*We assume that* $f^* \in \mathcal{F}_s$.

**Assumption 2** (Noise). *We assume the noise has zero mean, and its probability distribution has an exponentially decaying tail, i.e.,*

$$\mathbb{E}[\varepsilon] = 0, \qquad \mathbb{P}[|\varepsilon| > t] \leq c_0 e^{-\frac{t^2}{\sigma}} \; \forall t \geq \tau_0. \tag{4}$$

*Here* $c_0, \tau_0$ *and* $\sigma$ *are constants.*

The ultimate aim is to minimize the generalization error (expected risk) $L(\theta) = \mathbb{E}_{\mathbf{x}, y}[(f(\mathbf{x}; \theta) - y)^2]$. In practice, we only have at our disposal the empirical risk $\hat{L}(\theta) = \frac{1}{n} \sum_{i=1}^n (f(\mathbf{x}_i; \theta) - y_i)^2$. The generalization gap is defined as the difference between expected and empirical risk. We also define the truncated risks by $L_B(\theta) = \mathbb{E}_{\mathbf{x}, y}[(f(\mathbf{x}; \theta) - y)^2 \wedge B^2], \hat{L}_B(\theta) = \frac{1}{n} \sum_{i=1}^n (f(\mathbf{x}_i; \theta) - y_i)^2 \wedge B^2$.

## 2 PRELIMINARY

In this section, we summarize some results on the approximation error and generalization bound for two-layer ReLU networks, whose proofs are deferred to Appendix A and B. These results are required by our subsequent a priori analysis.

## 2.1 APPROXIMATION PROPERTIES

Most of the content is adapted from Barron (1993); Breiman (1993) and Klusowski & Barron (2016).

**Proposition 1.** *For any* $f \in \mathcal{F}_s$, *it has an integral representation as follows*

$$f(\mathbf{x}) - f(0) - \mathbf{x} \cdot \nabla f(0) = v \int_{\{-1,1\} \times [0,1] \times \mathbb{R}^d} h(\mathbf{x}; z, t, \boldsymbol{\omega}) dp(z, t, \boldsymbol{\omega}),$$

*where* $v < 2\gamma(f)$ *and*

$$s(z, t, \boldsymbol{\omega}) = -\text{sign}\big(\cos(\|\boldsymbol{\omega}\|_1 t - zb(\boldsymbol{\omega}))\big)$$
$$h(\mathbf{x}; z, t, \boldsymbol{\omega}) = s(z, t, \boldsymbol{\omega})\,(z\mathbf{x} \cdot \boldsymbol{\omega}/\|\boldsymbol{\omega}\|_1 - t)_+ .$$

For simplicity, in the rest of this paper, we assume $\nabla f(0) = 0$, $f(0) = 0$. We take $m$ samples $T_m = \{(z_1, t_1, \boldsymbol{\omega}_1), \ldots, (z_m, t_m, \boldsymbol{\omega}_m)\}$ with $(z_i, t_i, \boldsymbol{\omega}_i)$ randomly drawn from $p(z, t, \boldsymbol{\omega})$, and consider the empirical average $\hat{f}_m(\mathbf{x}) = \frac{v}{m} \sum_{k=1}^{m} h(\mathbf{x}; z_i, t_i, \boldsymbol{\omega}_i)$, which is exactly a two-layer ReLU network of width $m$. The central limit theorem (CLT) tells us that the approximation error is roughly

$$\mathbb{E}_{(z,t,\boldsymbol{\omega})}[h(\mathbf{x}; z, t, \boldsymbol{\omega})] - \frac{1}{m} \sum_{k=1}^{m} h(\mathbf{x}; z_k, t_k, \boldsymbol{\omega}_k) \approx \sqrt{\frac{\mathrm{Var}_{(z,t,\boldsymbol{\omega})}[h(\mathbf{x}; z, t, \boldsymbol{\omega})]}{m}}.$$

So as long as we can bound the variance at the right-hand side, we will have an estimate of the approximation error. The following result formalizes this intuition.

**Theorem 2.** *For any distribution $\pi$ with $\mathrm{supp}(\pi) \subset \Omega$ and any $f \in \mathcal{F}_s$, there exists a two-layer network $f(\mathbf{x}; \tilde{\theta})$ of width $m$ such that*

$$\mathbb{E}_{\mathbf{x} \sim \pi}|f(\mathbf{x}) - f(\mathbf{x}; \tilde{\theta})|^2 \leq \frac{16\gamma^2(f)}{m}.$$

*Furthermore $\|\tilde{\theta}\|_{\mathcal{P}} \leq 4\gamma(f)$, which means that the path norm of $\tilde{\theta}$ can be bounded by the spectral norm of the target function.*

## 2.2 ESTIMATING THE GENERALIZATION GAP

**Definition 3** (Rademacher complexity). Let $\mathcal{H}$ be a hypothesis space, i.e. a set of functions. The Rademacher complexity of $\mathcal{H}$ with respect to samples $S = (z_1, z_2, \ldots, z_n)$ is defined as $\hat{R}(\mathcal{H}) = \frac{1}{n}\mathbb{E}_\xi[\sup_{h \in \mathcal{H}} \sum_{i=1}^{n} h(z_i)\xi_i]$, where $\{\xi_i\}_{i=1}^{n}$ are independently random variables with $\mathbb{P}(\xi_i = +1) = \mathbb{P}(\xi_i = -1) = \frac{1}{2}$.

The generalization gap can be estimated via Rademacher complexity by the following theorem (see Bartlett & Mendelson (2002) and Shalev-Shwartz & Ben-David (2014) ).

**Theorem 3.** *Fix a hypothesis space $\mathcal{H}$, and suppose that for any $h \in \mathcal{H}$ and $z$, $|h(z)| \leq c$. Then for any $\delta > 0$, with probability at least $1 - \delta$ over the choice of $S = (z_1, z_2, \ldots, z_n)$, we have*

$$\sup_{h \in \mathcal{H}} |\frac{1}{n} \sum_{i=1}^{n} h(z_i) - \mathbb{E}_z[h(z)]| \leq 2\mathbb{E}_S[\hat{R}(\mathcal{H})] + c\sqrt{\frac{2\log(2/\delta)}{n}}.$$

About the Rademacher complexity of two-layer networks (1), we have the following result.

**Lemma 1.** *Let $z = (x, y)$ and $h(z; \theta) = \ell(f(x; \theta), y)$. Consider all the two-layer networks with path norm bounded by $Q$, i.e. $\mathcal{H}_Q := \{h(z; \theta) \mid \|\theta\|_{\mathcal{P}} \leq Q\}$. If loss function $\ell(y, \hat{y})$ is L-Lipschitz continuous with respect to $y$, then we have*

$$\hat{R}(\mathcal{H}_Q) \leq QL\sqrt{\frac{2\log(2d)}{n}}$$

Applying Theorem 3 and Lemma 1 gives us the following generalization bound.

**Theorem 4** (A posterior generalization bound). *For any $\delta > 0$, with probability at least $1 - \delta$ over the choice of the training set $S$, we have for any two-layer network $f(\mathbf{x}; \theta)$, the following result holds:*

$$|L_B(\theta) - \hat{L}_B(\theta)| \leq 4B(\|\theta\|_{\mathcal{P}} + 1)\sqrt{\frac{2\log(2d)}{n}} + B^2\sqrt{\frac{2\log(2c(1 + \|\theta\|_{\mathcal{P}})^2/\delta)}{n}}, \tag{5}$$

*where $c = \sum_{k=1}^{\infty} 1/k^2$.*

## 3 MAIN RESULTS

We see that the path norm of the special solution $\tilde{\theta}$ which achieves the optimal approximation error is independent of the network size, and this norm can also be used to bound the generalization gap (Theorem 4). Therefore, if the path norm is suitably penalized during training, we should be able to control the generalization gap without harming the approximation accuracy. One possible implementation of this idea is through the structural empirical risk minimization (Vapnik, 1998) as follows.

**Definition 4** (Path-norm regularized estimator). Let the path-norm regularized risk defined as

$$J_\lambda(\theta) := \hat{L}_{B_n}(\theta) + \lambda B_n \sqrt{\frac{2\log(2d)}{n}}(1 + \|\theta\|_\mathcal{P}), \tag{6}$$

where $B_n = 2 + \max\{\tau_0, \sigma\log n\}$ and $\lambda$ is a positive constant. The condition on $\lambda$ will be given below. The *path-norm regularized estimator* is defined as

$$\hat{\theta}_n = \arg\min J_\lambda(\theta). \tag{7}$$

It is worth noting that the minimizer is not necessarily unique, and $\hat{\theta}_n$ should be understood as any of the minimizers. About this estimator, we have the following result.

**Theorem 5** (Main Result). *Under Assumption 1 and 2, there exists a constant $C$ depending only on $\sigma, c_0$ such that for any $\delta > 0$ and $\lambda \geq 4$, with probability at least $1 - \delta$ over the choice of the training set $S$, the generalization error of estimator (7) satisfies*

$$\mathbb{E}|f(\mathbf{x}; \hat{\theta}_n) - f^*(\mathbf{x})|^2 \leq C\frac{\gamma^2(f^*)}{m} + C\frac{B_n^2}{\sqrt{n}}\left(\lambda\hat{\gamma}(f^*)\sqrt{\log(2d)} + \sqrt{\log(nc/\delta)}\right). \tag{8}$$

*Here $\hat{\gamma}(f^*) = \max\{\gamma(f^*), 1\}$.*

Since $B_n$ depends on the magnitude of noise, we actually prove

$$\mathbb{E}|f(\mathbf{x}; \hat{\theta}_n) - f^*(\mathbf{x})|^2 = \begin{cases} O(\frac{1}{m}) + O\left(\sqrt{\frac{\log d + \log n}{n}}\right) & \text{if } \sigma \leq \frac{C}{\log n} \\ O(\frac{1}{m}) + O\left(\log^2(n)\sqrt{\frac{\log d + \log n}{n}}\right) & \text{if } \sigma > \frac{C}{\log n}, \end{cases}$$

where $C$ is a constant. This means that the noise introduces at most an extra logarithmic term. Moreover, if the probability distribution function of the noise decays sufficiently fast (for example, bounded noise $\sigma = 0$), the logarithmic term can even be eliminated.

**Remark 1.** *It should be noted that both terms at the right hand side of the above result has a Monte Carlo nature, as can be seen later in the proof. From this viewpoint, the result is quite sharp. The dimensional dependence is mainly reflected in the norm $\gamma(f^*)$ (see Barron (1993)).*

**Comparison with existing results** Klusowski & Barron (2016) analyzed a similar problem. However they require the network width $m$ to be the orders of $\text{poly}(n)$. In contrast, our results allow the network width to be arbitrarily large. See Table 1 for the detailed comparison between our results and theirs.

| noise | zero | sub-Gaussian |
|---|---|---|
| Our results | $\frac{1}{m} + \left(\frac{\log d + \log n}{n}\right)^{1/2}$ | $\frac{1}{m} + \log^2(n)\left(\frac{\log d + \log n}{n}\right)^{1/2}$ |
| Results of Klusowski & Barron (2016) | $\left(\frac{\log d}{n}\right)^{1/3}$ | $\left(\frac{\log d}{n}\right)^{1/4}$ |

Table 1: Comparison between our work and Klusowski & Barron (2016).

## 4 PROOF OF MAIN RESULTS

### 4.1 NOISELESS CASE

For this case, we provide a rather short and intuitive sketch of proof, which helps to clarify the main idea of the complete proof in the next section. In the noiseless case, $\sigma = 0, \tau_0 = 0$, thus $B_n = 2$. The solution $\tilde{\theta}$ constructed in Theorem 2 satisfies $L(\tilde{\theta}) = O(m^{-1})$ and $\|\tilde{\theta}\|_\mathcal{P} = O(1)$. According to Theorem 4, we have $\hat{L}(\tilde{\theta}) = O(m^{-1}) + \tilde{O}(n^{-1/2})$ [1]. Hence the corresponding regularized risk satisfies

$$J_\lambda(\tilde{\theta}) = O(m^{-1}) + \tilde{O}(n^{-1/2}).$$

---

[1] Asymptotic notation $\tilde{O}(\cdot)$ is similar to $O(\cdot)$ but with logarithmic terms ignored.

By comparing the minimizer $\hat{\theta}_n$ with $\tilde{\theta}$, we have

$$J_\lambda(\hat{\theta}_n) \leq J_\lambda(\tilde{\theta}) = O(m^{-1}) + \tilde{O}(n^{-1/2}).$$

Furthermore $\|\hat{\theta}_n\|_\mathcal{P} \leq \tilde{O}(n^{1/2}m^{-1})$. By Theorem 4, we have

$$L(\hat{\theta}_n) \leq \hat{L}(\hat{\theta}_n) + 4\sqrt{\frac{2\log(2d)}{n}}(\|\hat{\theta}_n\|_\mathcal{P} + 1) + \tilde{O}\left(n^{-1/2}\sqrt{\log(\|\hat{\theta}_n\|_\mathcal{P})}\right). \tag{9}$$

As long as $\lambda \geq 4$, we have

$$L(\hat{\theta}_n) \leq J_\lambda(\hat{\theta}_n) + \tilde{O}\left(n^{-1/2}\sqrt{\log(\|\hat{\theta}_n\|_\mathcal{P})}\right) = O(m^{-1}) + \tilde{O}(n^{-1/2}).$$

We thus complete the proof.

This analysis highly relies on the fact that the approximation error and generalization gap can be controlled by the path norm simultaneously.

## 4.2 Noisy Case

In the presence of noise, the expected risk can be decomposed into three terms

$$\begin{aligned}
\mathbb{E}_{\mathbf{x},y}|f(x;\theta) - y|^2 &= \mathbb{E}_{\mathbf{x},\varepsilon}|f(x;\theta) - f^*(x) - \varepsilon|^2 \\
&= \mathbb{E}_{\mathbf{x},\varepsilon}|f(x;\theta) - f^*(x)|^2 + 2\mathbb{E}_{\mathbf{x},\varepsilon}[(f(x;\theta) - f^*(x))\varepsilon] + \mathbb{E}[\varepsilon^2].
\end{aligned}$$

Since $\varepsilon$ is independent of $\mathbf{x}$ and $\mathbb{E}[\varepsilon] = 0$, we have

$$L(\theta) = \mathbb{E}_{\mathbf{x}}|f(x;\theta) - f^*(x)|^2 + \mathbb{E}[\varepsilon^2].$$

This suggests that, in spite of noise, we still have

$$\operatorname{argmin}_\theta L(\theta) = \operatorname{argmin}_\theta \mathbb{E}_{\mathbf{x}}|f(x;\theta) - f^*(x)|^2, \tag{10}$$

and the latter is what we really want to minimize.

We first need to address the issue that $\hat{L}(\theta) - L(\theta)$ can be arbitrarily large, due to the presence of the noise. Let us consider the truncated risk $L_B(\theta) = \mathbb{E}[(f(x;\theta) - y)^2 \wedge B^2]$, which has the following property, whose proof is deferred to Appendix C.

**Lemma 2.** *Under Assumption 2, we have*

$$\sup_\theta |L(\theta) - L_{B_n}(\theta)| \leq \frac{2c_0\sigma^2}{\sqrt{n}},$$

By triangle inequality, we have $|L(\theta)| \leq |L(\theta) - L_{B_n}(\theta)| + |L_{B_n}(\theta)| = \frac{2c_0\sigma^2}{\sqrt{n}} + |\hat{L}_{B_n}(\theta)|$. Therefore this lemma tell us that as long as we can control the truncated risk, then the original risk will be controlled accordingly.

**Proposition 6.** *Let $\tilde{\theta}$ be the solution constructed in Theorem 2. Then with probability at least $1 - \delta$, we have*

$$J_\lambda(\tilde{\theta}) \leq L(\tilde{\theta}) + \frac{2c_0\sigma^2}{\sqrt{n}} + \frac{B_n^2}{\sqrt{n}}\left(\hat{\gamma}(f^*)\left(3 + 5(\lambda+4)\sqrt{\log(2d)}\right) + \sqrt{\log(2c/\delta)}\right),$$

*where $\hat{\gamma}(f) = \max\{\gamma(f), 1\}$.*

*Proof.* According to Definition 4 and the property that $\|\tilde{\theta}\|_\mathcal{P} \leq 4\gamma(f^*)$, the regularized cost of $\tilde{\theta}$ must satisfies

$$\begin{aligned}
J_\lambda(\tilde{\theta}) &= \hat{L}_{B_n}(\tilde{\theta}) + \lambda B_n\sqrt{\frac{2\log(2d)}{n}}(\|\tilde{\theta}\|_\mathcal{P} + 1) \\
&\overset{(1)}{\leq} L_{B_n}(\tilde{\theta}) + (4+\lambda)B_n\sqrt{\frac{2\log(2d)}{n}}(\|\tilde{\theta}\|_\mathcal{P} + 1) + B_n^2\sqrt{\frac{2\log(2c(1 + \|\tilde{\theta}\|_\mathcal{P})^2/\delta)}{n}} \\
&\overset{(2)}{\leq} L(\tilde{\theta}) + \frac{2c_0\sigma^2}{\sqrt{n}} + (\lambda+4)B_n\sqrt{\frac{2\log(2d)}{n}}(4\gamma(f^*) + 1) + B_n^2\sqrt{\frac{2\log(2c(1 + 4\gamma(f^*))^2/\delta)}{n}},
\end{aligned} \tag{11}$$

where $(1), (2)$ follow the Theorem 4 and Lemma 2, respectively. The last term can be simplified by using $\sqrt{a+b} \le \sqrt{a} + \sqrt{b}$ and $\log(1+a) \le a$ for $a \ge 0, b \ge 0$. So we have

$$
\begin{aligned}
\sqrt{2\log(2c(1+4\gamma(f^*))^2/\delta)} &\le \sqrt{2\log(2c/\delta)} + \sqrt{4\log(1+4\gamma(f^*))} \\
&\le 2\sqrt{\log(2c/\delta)} + 4\sqrt{\gamma(f^*)} \\
&\le 2\sqrt{\log(2c/\delta)} + 4\hat{\gamma}(f^*),
\end{aligned}
$$

where $\hat{\gamma}(f^*) = \max(\gamma(f^*), 1)$. By plugging it into Equation (11), and using $\hat{\gamma}(f^*) \ge 1$, $B_n \ge 1$, we have

$$
J_\lambda(\tilde{\theta}) \le L(\tilde{\theta}) + \frac{2c_0\sigma^2}{\sqrt{n}} + \frac{B_n^2}{\sqrt{n}}\left(\hat{\gamma}(f^*)\left(3 + 5(\lambda+4)\sqrt{\log(2d)}\right) + \sqrt{\log(2c/\delta)}\right).
$$

$\square$

**Proposition 7** (Properties of the regularized estimator). *The path-norm regularized estimator $\hat{\theta}_n$ satisfies:*

$$
J_\lambda(\hat{\theta}_n) \le J_\lambda(\tilde{\theta})
$$

$$
\|\hat{\theta}_n\|_{\mathcal{P}} \le \lambda^{-1}B_n^{-1}\sqrt{\frac{n}{2\log(2d)}}J_\lambda(\tilde{\theta})
$$

*Proof.* The first claim follows from the definition of $\hat{\theta}_n$. For the second claim, we have $\lambda\sqrt{\frac{\log(2d)}{n}}\|\hat{\theta}_n\|_{\mathcal{P}} \le J_\lambda(\hat{\theta}_n) \le J_\lambda(\tilde{\theta})$, so $\|\hat{\theta}_n\|_{\mathcal{P}} \le \lambda^{-1}B_n^{-1}\sqrt{\frac{n}{2\log(2d)}}J_\lambda(\tilde{\theta})$. $\square$

**Remark 2.** *The above proposition establishes the connection between the regularized solution and the special solution $\tilde{\theta}$ constructed in Theorem 2. According to Proposition 6, we can conclude that the upper bound of the generalization gap satisfies $\frac{\|\hat{\theta}_n\|_{\mathcal{P}}}{\sqrt{n}} = O(L(\tilde{\theta})) + \tilde{O}(\gamma(f^*)n^{-1/2}) \to O(L(\tilde{\theta}))$, as $n \to \infty$. It suggests that our regularization term is added appropriately, which forces the generalization gap to be roughly in the same order of approximation error.*

**Proof of Theorem 5.** We are now ready to prove our main theorem. Let $C_1 = 2c_0\sigma^2$. Lemma 2 implies that $L(\hat{\theta}_n) \le L_{B_n}(\hat{\theta}_n) + \frac{C_1}{\sqrt{n}}$. Then we have

$$
\begin{aligned}
L(\hat{\theta}_n) &\overset{(1)}{\le} \hat{L}_{B_n}(\hat{\theta}_n) + 4B_n(\|\hat{\theta}_n\|_{\mathcal{P}} + 1)\sqrt{\frac{2\log(2d)}{n}} + B_n^2\sqrt{\frac{\log(2c(1+\|\hat{\theta}_n\|_{\mathcal{P}})^2/\delta)}{n}} + \frac{C_1}{\sqrt{n}} \\
&\overset{(2)}{\le} J_\lambda(\hat{\theta}_n) + B_n^2\sqrt{\frac{\log(2c(1+\|\hat{\theta}_n\|_{\mathcal{P}})^2/\delta)}{n}} + \frac{C_1}{\sqrt{n}}
\end{aligned}
$$

Where (1) follows from the a posteriori generalization bound in Theorem 4, and (3) is due to $\lambda \ge 4$. Furthermore,

$$
\begin{aligned}
\sqrt{\log(2c(1+\|\hat{\theta}_n\|_{\mathcal{P}})^2/\delta)} &\le \sqrt{\log(2nc/\delta)} + \sqrt{2\log(1+n^{-1/2}\|\hat{\theta}_n\|_{\mathcal{P}})} \\
&\le \sqrt{\log(2nc/\delta)} + \sqrt{2n^{-1/2}\|\hat{\theta}_n\|_{\mathcal{P}}}.
\end{aligned}
$$

By plugging it back and simplifying the right hand side according to Proposition 6 and Proposition 7, we conclude that there exists a constant $C_2$ such that

$$
L(\hat{\theta}_n) \le L(\tilde{\theta}) + C_2\frac{B_n^2}{\sqrt{n}}\left(\lambda\hat{\gamma}(f^*)\sqrt{\log(2d)} + \sqrt{\log(nc/\delta)}\right).
$$

By applying the decomposition that $L(\theta) = \mathbb{E}|f(\mathbf{x};\theta) - f^*(\mathbf{x})|^2 + \mathbb{E}[\varepsilon^2]$, and the result of Theorem 2, we obtain

$$
\mathbb{E}|f(\mathbf{x};\hat{\theta}_m) - f^*(\mathbf{x})|^2 \le C\frac{\gamma^2(f^*)}{m} + C\frac{B_n^2}{\sqrt{n}}\left(\lambda\hat{\gamma}(f^*)\sqrt{\log(2d)} + \sqrt{\log(nc/\delta)}\right).
$$

Here the constant $C$ depends only on $\sigma, c_0$.

**Remark 3.** *From the proof, we can see that the requirement of $\lambda \ge 4$ is due to constant 4 appears in the upper bound of the generalization gap. If we have a sharper generalization bound, then $\lambda$ could be set smaller.*

## 5 NUMERICAL EXPERIMENTS

We evaluate the properties of the regularized estimator on both MNIST[2] (LeCun et al., 1998) and CIFAR-10[3] (Krizhevsky & Hinton, 2009) datasets. Each example in MNIST is a $28 \times 28$ grayscale image, while each example in CIFAR-10 is a $32 \times 32 \times 3$ color image. To be consistent with our setup in theoretical analysis, we restrict ourselves to a binary classification problem. For MNIST, we map numbers $\{0, 1, 2, 3, 4\}$ to label 0 and $\{5, 6, 7, 8, 9\}$ to 1. For CIFAR-10, we select the examples with labels 0 and 1 to construct our new training and validation sets. Thus, our new MNIST has $60,000$ training examples, and CIFAR-10 has $10,000$ training examples. The mean squared error rather than cross entropy is used as our loss function.

Following the standard strategy (He et al., 2015), the two-layer ReLU network is initialized using $a_i \sim \mathcal{N}(0, \frac{2\kappa}{m})$, $b_{i,j} \sim \mathcal{N}(0, 2\kappa/d)$, $c_i = 0$. We use $\kappa = 1$ and train the models using the Adam optimizer (Kingma & Ba, 2015) for $T = 10,000$ steps, unless it is specified otherwise. The initial learning rate is set to be $0.001$, and it is then multiplied by a decay factor of $0.1$ at $0.7T$ and again at $0.9T$. We set the trade-off parameter $\lambda = 0.1$ for regularized models. Although the theoretical results suggest $\lambda \geq 4$, we find in practice usually a smaller $\lambda$ can achieve better test performance.

### 5.1 THE NON-VACUOUS UPPER BOUND OF THE GENERALIZATION GAP

Theorem 4 shows that the generalization gap can be bounded by $\frac{\|\theta\|_{\mathcal{P}}}{\sqrt{n}}$ up to some constants. To see how this works in practice, we trained both regularized models with $\lambda = 0.1$ and un-regularized models ($\lambda = 0$) for fixed network width $m = 10,000$. To cover the over-parameterization regime, we also consider $n = 100$ where $m/n = 100 \gg 1$. The results are summarized in Table 2.

Table 2: Comparison of regularized ($\lambda = 0.1$) and un-regularized ($\lambda = 0$) models. The experiments are repeated for 5 times, and the mean values are reported.

| dataset | $\lambda$ | n | training accuracy | test accuracy | $\frac{\|\theta\|_{\mathcal{P}}}{\sqrt{n}}$ |
|---|---|---|---|---|---|
| CIFAR-10 | 0 | $10^4$ | 100% | 84.5% | 58 |
| | | 100 | 100% | 70.5% | 507 |
| | 0.1 | $10^4$ | 87.4% | 86.9% | **0.14** |
| | | 100 | 91.0% | 72.0% | **0.43** |
| MNIST | 0 | $6 \times 10^4$ | 100% | 98.8% | 58 |
| | | 100 | 100% | 78.7% | 162 |
| | 0.1 | $6 \times 10^4$ | 98.1% | 97.8% | **0.27** |
| | | 100 | 100% | 74.9% | **0.41** |

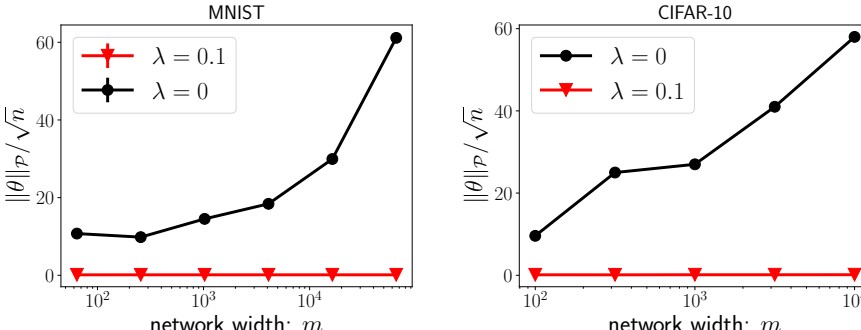

Figure 1: Comparison of path norms between regularized and un-regularized solutions for varying widths.

As we can see, the test accuracies of regularized and un-regularized solutions are generally comparable, but the upper bounds of generalization gap $\frac{\|\theta\|_{\mathcal{P}}}{\sqrt{n}}$ are dramatically different. Specifically, for un-regularized models, the bounds are always vacuous, since they are several orders of magnitude larger than the naively upper bound 1. This is consistent with the observations in Arora et al. (2018)

---

[2] http://yann.lecun.com/exdb/mnist/
[3] https://www.cs.toronto.edu/~kriz/cifar.html

and Neyshabur et al. (2018b). However, for regularized models, the bounds are non-vacuous, although they are still far from the true values. These numerical observations are consistent with our theoretical prediction in Proposition 7.

To further explore the impact of over-parameterization, we trained various models with different widths. For both datasets, all the training examples are used. In Figure 1, we display how the upper bound $\frac{\|\theta\|_{\mathcal{P}}}{\sqrt{n}}$ of the learned solution varies with the network width. We find that this quantity for the regularized model is almost constant, whereas for the original model it increases with network width. This provides numerical evidence that our theoretical results hold for the network with an arbitrary width.

## 5.2 Dependence on the Initialization

Since the neural network model is non-convex, it is interesting to see how the initialization affects the performance of the solutions, especially in the over-parametrized regime. To this end, we fix $m = 10000, n = 100$ and vary the variance of random initialization $\kappa$. The results are reported in Figure 2. In general, we find that regularized models are much more stable than the un-regularized models. For large initialization, the regularized model always performs significantly better.

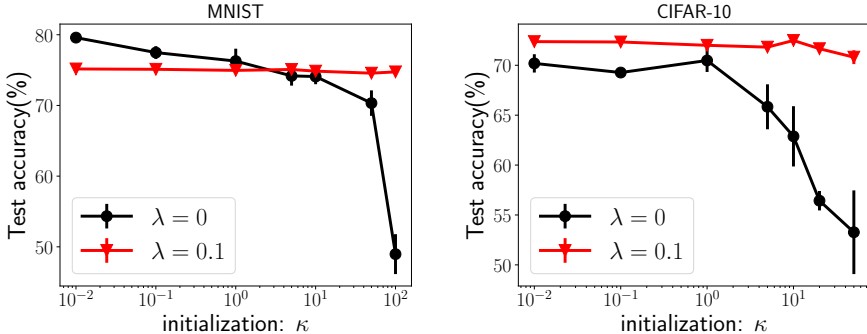

Figure 2: Test accuracies of solutions obtained from different initializations. Each experiment is repeated for 5 times, and we report the mean and standard deviation.

## 6 Concluding Remarks

The most unsatisfactory aspect of our result is that it is proved for the regularized model. In practice it is uncommon to add explicit regularizations. Instead, practitioners rely on the so-called "implicit regularization" (Zhang et al., 2017; Neyshabur, 2017). At the moment it is unclear where the "implicit regularization" comes from and how it actually works. But there are overwhelming evidence that by tuning extensively the details of the optimization procedure, including the algorithm, the initialization, the hyper-parameters, etc., one can find solutions with superior performance on the test data. The disadvantage is that excessive tuning and serious experience is required to find good solutions. Until we have a good understanding about the mysteries surrounding implicit regularization, the business of parameter tuning will remain an art. In contrast, the regularized model is rather robust and much more fool-proof. Borrowing the terminology from mathematical physics, we consider the regularized model to be "well-posed" and the original model to be "ill-posed" .

There are two clear paths moving forward. One is to study other regularized models. In fact to avoid the slight loss of test accuracy shown for the MNIST dataset in Figure 1, one can consider regularizations that vanish for small values of the path norm. Our main results should hold for this kind of regularizations. The other is to study the so-called "implicit regularization". Recently, assuming that the data is well-separated, Brutzkus et al. (2018); Li & Liang (2018) proved that for two-layer networks, the number of iterations required for SGD to achieve certain accuracy for the classification problem is independent of the network size. Implicit regularization has also been studied in other problems, such as logistic regression (Soudry et al., 2018) and matrix factorization (Li et al., 2018; Gunasekar et al., 2017).

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

## A    PROOFS FOR APPROXIMATION PROPERTIES

**Proof of Proposition 1**    By an abuse of notation, let $f$ be its own $L^2$ extension in $\mathbb{R}^d$. Since $f \in L^2(\mathbb{R}^d)$, $f(\mathbf{x}) - \mathbf{x} \cdot \nabla f(0) - f(0)$ can be written as

$$\int_{\mathbb{R}^d} \left( e^{i\boldsymbol{\omega} \cdot \mathbf{x}} - i\boldsymbol{\omega} \cdot \mathbf{x} - 1 \right) \hat{f}(\boldsymbol{\omega}) d\boldsymbol{\omega}. \tag{12}$$

Note that the following identity

$$-\int_0^c \left[ (z-s)_+ e^{is} + (-z-s)_+ e^{-is} \right] ds = e^{iz} - iz - 1$$

holds when $|z| \leq c$. Choosing $c = \|\boldsymbol{\omega}\|_1$, $z = \boldsymbol{\omega} \cdot \mathbf{x}$, we have

$$|z| \leq \|\boldsymbol{\omega}\|_1 \|\mathbf{x}\|_\infty \leq c.$$

Let $s = \|\boldsymbol{\omega}\|_1 t$, $0 \leq t \leq 1$, and $\hat{\boldsymbol{\omega}} = \boldsymbol{\omega}/\|\boldsymbol{\omega}\|_1$, we have

$$-\|\boldsymbol{\omega}\|_1^2 \int_0^1 \left[ (\hat{\boldsymbol{\omega}} \cdot \mathbf{x} - t)_+ e^{i\|\boldsymbol{\omega}\|_1 t} + (-\hat{\boldsymbol{\omega}} \cdot \mathbf{x} - t)_+ e^{-i\|\boldsymbol{\omega}\|_1 t} \right] dt = e^{i\boldsymbol{\omega} \cdot \mathbf{x}} - i\boldsymbol{\omega} \cdot \mathbf{x} - 1. \tag{13}$$

Let $\hat{f}(\boldsymbol{\omega}) = e^{ib(\boldsymbol{\omega})} |\hat{f}(\boldsymbol{\omega})|$, inserting (13) into (12) yields

$$f(\mathbf{x}) - \mathbf{x} \cdot \nabla f(0) - f(0) = \int_{\mathbb{R}^d} \int_0^1 g(t, \boldsymbol{\omega}) dt d\boldsymbol{\omega},$$

where

$$g(t, \boldsymbol{\omega}) = -\|\boldsymbol{\omega}\|_1^2 |\hat{f}(\boldsymbol{\omega})| \left[ (\hat{\boldsymbol{\omega}} \cdot \mathbf{x} - t)_+ \cos(\|\boldsymbol{\omega}\|_1 t + b(\boldsymbol{\omega})) + (-\hat{\boldsymbol{\omega}} \cdot \mathbf{x} - t)_+ \cos(\|\boldsymbol{\omega}\|_1 t - b(\boldsymbol{\omega})) \right].$$

Consider a density on $\{0, 1\} \times [0, 1] \times \mathbb{R}^d$ defined by

$$p(z, t, \boldsymbol{\omega}) = |\hat{f}(\boldsymbol{\omega})| \|\boldsymbol{\omega}\|_1^2 |\cos(\|\boldsymbol{\omega}\|_1 t - zb(\boldsymbol{\omega}))| / v \tag{14}$$

where the normalized constant $v$ is given by

$$v = \int_{\mathbb{R}^d} \int_0^1 |\hat{f}(\boldsymbol{\omega})| \|\boldsymbol{\omega}\|_1^2 \left( |\cos(\|\boldsymbol{\omega}\|_1 t + b(\boldsymbol{\omega}))| + |\cos(\|\boldsymbol{\omega}\|_1 t - b(\boldsymbol{\omega}))| \right) d\boldsymbol{\omega} dt. \tag{15}$$

Since $f$ belongs to $\mathcal{F}_s$, so we have

$$v \leq 2\gamma(f) < +\infty, \tag{16}$$

therefore the density $p(z, t, \boldsymbol{\omega})$ is well-defined. To simplify the notations, we let

$$s(z, t, \boldsymbol{\omega}) = -\text{sign}\left( \cos(\|\boldsymbol{\omega}\|_1 t - zb(\boldsymbol{\omega})) \right) \tag{17}$$

$$h(\mathbf{x}; z, t, \boldsymbol{\omega}) = s(z, t, \boldsymbol{\omega}) \left( z\hat{\boldsymbol{\omega}} \cdot \mathbf{x} - t \right)_+ . \tag{18}$$

Then we have

$$f(\mathbf{x}) - \mathbf{x} \cdot \nabla f(0) - f(0) = v \int_{\{-1,1\} \times [0,B] \times \mathbb{R}^d} h(\mathbf{x}; z, t, \boldsymbol{\omega}) dp(z, t, \boldsymbol{\omega}). \tag{19}$$

Since $\mathbf{x} = (\mathbf{x})_+ - (-\mathbf{x})_+$, we obtain

$$f(\mathbf{x}) = f(0) + (\mathbf{x} \cdot \nabla f(0))_+ - (-\mathbf{x} \cdot \nabla f(0))_+ + v \int_{\{-1,1\} \times [0,B] \times \mathbb{R}^d} h(\mathbf{x}; z, t, \boldsymbol{\omega}) dp(z, t, \boldsymbol{\omega}).$$

Therefore $f \in \bar{\mathcal{H}}_\sigma$.

**Proof of Theorem 2** Let $\hat{f}_m(\mathbf{x}) = \frac{v}{m}\sum_{k=1}^{m} h(\mathbf{x}; z_i, t_i, \boldsymbol{\omega}_i)$ be the Monte-Carlo estimator, we have

$$\mathbb{E}_{T_m}\mathbb{E}_{\mathbf{x}}|f(\mathbf{x}) - \hat{f}_m(\mathbf{x})|^2 = \mathbb{E}_{\mathbf{x}}\mathbb{E}_{T_m}|f(\mathbf{x}) - \hat{f}_m(\mathbf{x})|^2$$
$$= \frac{v^2}{m}\mathbb{E}_{\mathbf{x}}\left(\mathbb{E}_{(z,t,\boldsymbol{\omega})}[h^2(\mathbf{x}; z, t, \boldsymbol{\omega})] - f^2(\mathbf{x})\right)$$
$$\leq \frac{v^2}{m}\mathbb{E}_{\mathbf{x}}\mathbb{E}_{(z,t,\boldsymbol{\omega})}[h^2(\mathbf{x}; z, t, \boldsymbol{\omega})]$$

Furthermore, for any fixed $\mathbf{x}$, the variance can be upper bounded since

$$\mathbb{E}_{(z,t,\boldsymbol{\omega})}[h^2(\mathbf{x}; z, t, \boldsymbol{\omega})] \leq \mathbb{E}_{(z,t,\boldsymbol{\omega})}\left[(z\hat{\boldsymbol{\omega}} \cdot \mathbf{x} - t)_+^2\right]$$
$$\leq \mathbb{E}_{(z,t,\boldsymbol{\omega})}\left[(|\hat{\boldsymbol{\omega}} \cdot \mathbf{x}| + t)^2\right]$$
$$\leq 4.$$

Hence we have

$$\mathbb{E}_{T_m}\mathbb{E}_{\mathbf{x}}|f(\mathbf{x}) - \hat{f}_m(\mathbf{x})|^2 \leq \frac{4v^2}{m} \leq \frac{16\gamma^2(f)}{m}$$

Therefore there must exist a set of $T_m$, such that the corresponding empirical average satisfies

$$\mathbb{E}_{\mathbf{x}}|f - f_m|^2 \leq \frac{16\gamma^2(f)}{m}.$$

Due to the special structure of the Monte-Carlo estimator, we have $|a_k| = v/m, \|\boldsymbol{b}_k\|_1 = 1, |c_k| \leq 1$. It follows Equation (16) that $\|\tilde{\theta}\|_{\mathcal{P}} \leq 2v \leq 4\gamma(f)$.

# B    PROOFS FOR GENERALIZATION BOUNDS

Before to provide the upper bound for the Rademacher complexity of two-layer networks, we first need the following two lemmas.

**Lemma 3** (Lemma 26.11 of Shalev-Shwartz & Ben-David (2014)). *Let $S = (\mathbf{x}_1, \ldots, \mathbf{x}_n)$ be $n$ vectors in $\mathbb{R}^d$. Then the Rademacher complexity of $\mathcal{H}_1 = \{\mathbf{x} \mapsto \boldsymbol{u} \cdot \mathbf{x} \mid \|\boldsymbol{u}\|_1 \leq 1\}$ has the following upper bound,*

$$\hat{R}(\mathcal{H}_1) \leq \max_i \|\mathbf{x}_i\|_\infty \sqrt{\frac{2\log(2d)}{n}}$$

The above lemma characterizes the Rademacher complexity of a linear predictor with $\ell_1$ norm bounded by 1. To handle the influence of nonlinear activation function, we need the following contraction lemma.

**Lemma 4** (Lemma 26.9 of Shalev-Shwartz & Ben-David (2014)). *Let $\phi_i : \mathbb{R} \mapsto \mathbb{R}$ be a $\rho-$Lipschitz function, i.e. for all $\alpha, \beta \in \mathbb{R}$ we have $|\phi_i(\alpha) - \phi_i(\beta)| \leq \rho|\alpha - \beta|$. For any $\boldsymbol{a} \in \mathbb{R}^n$, let $\boldsymbol{\phi}(\boldsymbol{a}) = (\phi_1(a_1), \ldots, \phi_n(a_n))$, then we have*

$$\hat{R}(\boldsymbol{\phi} \circ \mathcal{H}) \leq \rho\hat{R}(\mathcal{H})$$

We are now ready to characterize the Rademacher complexity of two-layer networks. We use the path norm to control the complexity of the network.

**Lemma 5.** *Let $\mathcal{F}_Q = \{f_m(x; \theta) \mid \|\theta\|_{\mathcal{P}} \leq Q\}$ be the set of two-layer networks with path norm bounded by $Q$, then we have*

$$\hat{R}(\mathcal{F}_Q) \leq Q\sqrt{\frac{2\log(2d)}{n}}$$

*Proof.* To simplify the proof, we let $c_k = 0$, otherwise we can define $\boldsymbol{b}_k = (\boldsymbol{b}_k^T, c_k)^T$ and $\mathbf{x} = (\mathbf{x}^T, 1)^T$.

$$
\begin{aligned}
n\hat{R}(\mathcal{F}_Q) &= \mathbb{E}_\xi\Big[\sup_{\|\theta\|_\mathcal{P}\leq Q}\sum_{i=1}^n \xi_i \sum_{k=1}^m a_k\|\boldsymbol{b}_k\|_1\sigma(\hat{\boldsymbol{b}}_k^T\mathbf{x}_i)\Big] \\
&\leq \mathbb{E}_\xi\Big[\sup_{\|\theta\|_\mathcal{P}\leq Q,\|\boldsymbol{u}_k\|_1=1}\sum_{i=1}^n \xi_i \sum_{k=1}^m a_k\|\boldsymbol{b}_k\|_1\sigma(\boldsymbol{u}_k^T\mathbf{x}_i)\Big] \\
&= \mathbb{E}_\xi\Big[\sup_{\|\theta\|_\mathcal{P}\leq Q,\|\boldsymbol{u}_k\|_1=1}\sum_{k=1}^m a_k\|\boldsymbol{b}_k\|_1 \sum_{i=1}^n \xi_i\sigma(\boldsymbol{u}_k^T\mathbf{x}_i)\Big] \\
&\leq \mathbb{E}_\xi\Big[\sup_{\|\theta\|_\mathcal{P}\leq Q}\sum_{k=1}^m |a_k\|\boldsymbol{b}_k\|_1| \sup_{\|\boldsymbol{u}\|_1=1}|\sum_{i=1}^n \xi_i\sigma(\boldsymbol{u}^T\mathbf{x}_i)|\Big] \\
&\leq Q\mathbb{E}_\xi\Big[\sup_{\|\boldsymbol{u}\|_1=1}|\sum_{i=1}^n \xi_i\sigma(\boldsymbol{u}^T\mathbf{x}_i)|\Big] \leq Q\mathbb{E}_\xi\Big[\sup_{\|\boldsymbol{u}\|_1\leq 1}|\sum_{i=1}^n \xi_i\sigma(\boldsymbol{u}^T\mathbf{x}_i)|\Big] \\
&= Q\mathbb{E}_\xi\Big[\sup_{\|\boldsymbol{u}\|_1\leq 1}\sum_{i=1}^n \xi_i\sigma(\boldsymbol{u}^T\mathbf{x}_i)\Big]
\end{aligned}
$$

Since $\sigma$ is a $1-$Lipschitz continuous, by applying Lemma 4 and Lemma 3, we obtain

$$
\hat{R}(\mathcal{F}_Q) \leq Q\sqrt{\frac{2\log(2d)}{n}}.
$$

$\square$

**Proof of Lemma 1**  Since for any $y_i$, $\ell(y, y_i) = (y - y_i)^2 \wedge B^2$ is $2B-$Lipschitz continuous, by applying the contraction property of Rademacher complexity and Lemma 5, for $\mathcal{H}_Q = \{\ell \circ f \,|\, f \in \mathcal{F}_Q\}$ we have

$$
\hat{R}(\mathcal{H}_Q) \leq 2BQ\sqrt{\frac{2\log(2d)}{n}}.
$$

Directly applying Theorem 3 yields the result.

**Proposition 8.** *For the truncated risk, we have, with probability at least $1 - \delta$,*

$$
\sup_{\|\theta\|_\mathcal{P}\leq Q} |\hat{G}_B(\theta)| \leq 4BQ\sqrt{\frac{2\log(2d)}{n}} + B^2\sqrt{\frac{2\log(2/\delta)}{n}} \tag{20}
$$

**Proof of Theorem 4**  Consider the decomposition $\mathcal{F} = \cup_{l=1}^\infty \mathcal{F}_l$, where $\mathcal{F}_l = \{f_m(\mathbf{x};\theta)\,|\,\|\theta\|_\mathcal{P} \leq l\}$. Let $\delta_l = \frac{\delta}{cl^2}$ where $c = \sum_{l=1}^\infty \frac{1}{l^2}$. According to Theorem 8, if we fixed $l$ in advance, then with probability at least $1 - \delta_l$ over the choice of $S$,

$$
\sup_{\|\theta\|_\mathcal{P}\leq l} |G_n(\theta)| \leq 4Bl\sqrt{\frac{2\log(2d)}{n}} + B^2\sqrt{\frac{2\log(2/\delta_l)}{n}}.
$$

So the probability that there exists at least one $l$ such that (B) fails is at most $\sum_{l=1}^\infty \delta_l = \delta$. In other words, with probability at least $1 - \delta$, the inequality (B) holds for all $l$.

Given an arbitrary set of parameters $\theta$, denote $l_0 = \min\{l\,|\,\|\theta\|_\mathcal{P} \leq l\}$, then $l_0 \leq \|\theta\|_\mathcal{P} + 1$. Equation (B) implies that

$$
\begin{aligned}
|G_n(\theta)| &\leq 4Bl_0\sqrt{\frac{2\log(2d)}{n}} + B^2\sqrt{\frac{2\log(2cl_0^2/\delta)}{n}} \\
&\leq 4B(\|\theta\|_\mathcal{P} + 1)\sqrt{\frac{2\log(2d)}{n}} + B^2\sqrt{\frac{2\log(2c(1 + \|\theta\|_\mathcal{P})^2/\delta)}{n}}.
\end{aligned}
$$

## C  THE POOF OF LEMMA 2

*Proof.* Let $Z = f_m(\mathbf{x}; \theta) - f^*(\mathbf{x}) - \varepsilon$, then for any $B \geq 2 + \tau_0$, we have

$$
\begin{aligned}
|L(\theta) - L_B(\theta)| &= \mathbb{E}\left[(Z^2 - B^2)\mathbf{1}_{|Z| \geq B}\right] \\
&= \int_0^\infty \mathbb{P}\{Z^2 - B^2 \geq t^2\}dt^2 \leq \int_0^\infty \mathbb{P}\{|Z| \geq \sqrt{B^2 + t^2}\}dt^2 \\
&\leq \int_0^\infty \mathbb{P}\{|\varepsilon| \geq \sqrt{B^2 + t^2} - 2\}dt^2 \\
&= c_0 \int_B^\infty e^{-\frac{s^2}{2\sigma^2}} ds^2 = 2c_0\sigma^2 \int_{B^2/2\sigma^2}^\infty e^{-s}ds \\
&= 2c_0\sigma^2 e^{-B^2/2\sigma^2}
\end{aligned}
$$

Since $B_n \geq 2 + \max\{\tau_0, \sigma^2 \log n\}$, we have $2c_0\sigma^2 e^{-\frac{B_n^2}{2\sigma^2}} \leq 2c_0\sigma^2 n^{-1/2}$. Therefore,

$$
\sup_\theta |L(\theta) - L_{B_n}(\theta)|] \leq \frac{2c_0\sigma^2}{\sqrt{n}}.
$$

$\square$

