# OpenReview forum: "A Priori Estimates  of the Generalization Error for Two-layer Neural Networks"
_ICLR.cc/2019/Conference_

### Official Review · AnonReviewer3 · 2018-11-02
**Nice Idea; seems a bit incremental over known results**

**Rating:** 5
**Confidence:** 3

**Review:**

This paper studies the generalization properties of a two layer neural network for a nonlinear regression problem where the target function has a finite spectral norm. The generalization bound comprises of an approximation term (dependent on the width) and an estimation term (dependent on spectral norm of the target and scaling as 1/sqrt{n}).

The key contribution is the derivation of the generalization bound where the estimation term depends on the properties of the target function rather than properties of the class of two layer neural networks. These bounds are instantiated for a class of regularized estimators (with path norm regularization).

1. Theoretical Novelty: While both Theorem 3 (approximation) and Theorem 4 (a posterior generalization) were mostly following known results, the key development seems to be the bound on the path norm of the regularized solution in terms of the spectral norm of the target function. Given that the estimator is a path-norm regularized estimator, this seemed to be an incremental contribution. What would be more interesting is to obtain such a bound for an unregularized estimator: either saying something about the optimization procedure or relating this kind of regularization to properties of the dataset over which it is trained.

2. Regression vs Classification: While the focus of the paper is on a regression problem, the experiments and problem motivation seems to arise from a classification setting. This creates a mismatch between the what the paper is about and the problem that has been motivated. Would it be possible to extend these results to loss functions (other than squared loss ) like cross-entropy loss or hinge loss which indeed work in the classification setting?

3. Comparison with Klusowski & Barron (2016): In the comparison section, it is mentioned that  Klusowski & Barron (2016) analyze a "similar" problem and obtain worse generalization bounds. It would be important to know the exact setting in which they obtained their bounds and how do their assumptions compare with the ones made in this paper. The comparison seems incomplete without this.

4. The experiments showcase that the regularized estimator has a better path norm (and expectedly so) but almost similar (in case of MNIST actually better) test accuracy. This defeats the purpose of  showing that the regularized estimator has better generalization ability which is claimed in the introduction as well as the experiment section (calling it "well-posed"). What this indeed shows is that even though the path norm might be big, the generalization of the estimator is till very good contradicting the statements made.

5. The numbers shown in Figure 1 and the numbers reported in Table 2 do no match: while the plot shows that the scaled path norm is around 60 for both MNIST and CIFAR-10, the corresponding numbers in the table are 507 and 162. Can you please point out the reason for this discrepancy?

6. Theorem 5 seems to suggest that in the noiseless case, the estimation error would scale as the spectral norm of f^*. Rather, in the noiseless setting, it seems that the correct scaling of the generalization error should be with respect to properties of the regularized estimator and the function class. Even though the spectral norm can be arbitrarily high, the generalization bound should only be dependent on the complexity of functions which can fit the current data well. It would be good to have a comment in the draft on why the current dependence is a better thing and examples where such generalization bounds are indeed better.

---

### Official Review · AnonReviewer1 · 2018-11-07
**Incremental work from previous results**

**Rating:** 4
**Confidence:** 4

**Review:**

The main contribution of the paper is claimed as providing “apriori” guarantees for generalization where the generalization bounds depend only on the norm of the “true predictor”. This in contrast to what is termed as “posterior” guarantee where the generalization bound is provided in terms of the “learned predictor”.

I could not appreciate the main motivation/challenge in this distinction. In particular, if additional constraints are imposed on the learned predictor during training, then it seems straightforward that any “posterior” guarantee can be immediately converted to an “apriori” guarantee. For example, in the context of the problem in this paper, since it is known that the true function f* is approximated by a 2 layer network of path norm < 4\gamma(f*). Thus,  if during training the empirical loss is minimized over the networks with hard constraint of $\theta_P < 4\gamma(f*)$, then from Lemma 2, we should immediately get a generalization bound in terms of $\gamma(f*)$ instead of $||\theta||_P$.

The main result in the paper (Theorem 5) seems to essentially do a variant of the above approach, except instead of a hard constraint on the path norm of parameters, the authors analyze the estimator from soft regularization (eq. 6) and provide the guarantee in terms of the regularization parameter. This although is not immediate like the hard constrained version, I do not see major challenges in the extension to regularized version. Moreover, I also do not see the immediate motivation for analyzing the regularized objective over the constrained one (at least for the purpose of generalization bounds since both estimators are intractable computationally). Please provide a response if I missed something.


Writing:
1 (MAIN). Proper citations missing from many places. e.g., definition of path norm, spectral norm.
More importantly for the theorems in Section 2, appropriate citations within the theorems are missing. For example in Theorem 2 the authors do cite Klusowski and Barron in the beginning of section but not in the theorem header. Same for Theorem 3, and 4 which are known results too. Lemma 1 follows from corollary 7 in Neyshabur et al. 2015.


2. Undefined notation ||.||_\infty in Assumption 1.
3. Definition 2: please provide proper reference for the definition and for multivariate Fourier transform.

More generally it is hard to follow the notations through out the paper and the motivations for certain notations are not completely clear. Some intuition here would help. For example, the reason we care about Assumption 1 is that this class of functions are well approximated by 2 layer networks (as shown in theorem 2). This could be stated ahead of the definitions.


 Also a broader outline of the structure should help with readablity.

---

### Official Review · AnonReviewer2 · 2018-11-13
**Incremental results using known analysis tools**

**Rating:** 4
**Confidence:** 3

**Review:**

This paper provides a new generalization bound for two layer neural network with regularization. The main analysis tool is to bound the Rademacher complexity of the network (due to regularization). While the work achieves a bound that is superior than a previous work, I personally find the work less inspiring and somewhat incremental. I have three main concerns of the result/work:

1. The analysis is on a very shallow network. It is not clear how this result shed insight on understanding the success of *deep* neural network.

2. The work is restricted to analyzing a NN with explicit regularization. As the authors noted themselves, such a paradigm is less popular in practice now.

3. The analysis tool - bounding Rademacher complexity - is very standard

---

### Official Review · AnonReviewer4 · 2018-11-13
**Interesting ideas but unclear how they contribute to a better understanding of generalization of neural networks**

**Rating:** 4
**Confidence:** 3

**Review:**

The authors consider the notion of path norm for two layer ReLu network, and derive a generalization bound for a path-norm regularized estimator under a regression model.

I apologize for the cursory review, as I have only been asked to review this paper two days ago. I have two main concerns about this paper: it is not clear to me how the distinction between “a priori” and “a posteriori” estimates enables a better understanding of the problem of generalization, and it is not clear to me how this paper contributes to a better understanding of generalization for neural networks.

A priori v. a posteriori:

The authors attempt to distinguish “a priori” and “a posteriori” bounds by whether they depend on the true or estimated regression function. However, note that “a priori bounds” are significantly different from the common meaning of a “generalization bound”, which is most often understood to obtain a model-free bound of the generalization error. I found the reference to (Dziugaite and Roy, 2017) particularly confusing, as I am not sure how a PAC-Bayesian approach corresponds to an “a priori” approach.

Additionally, it seems that the theorems in the paper are phrased in the realizable case (i.e. where the true data-generating distribution is part of the model class). I believe that this is a poor model in the context of neural networks, which are often used to approximate complex models. Indeed, the authors claim that: “if the path-norm is suitably penalized during training, we should be able to control the generalization gap without harming approximation accuracy” which is true in the realizable case. However, the authors’ experiment (table 2) show that the training (and testing!) performance tends to decrease when regularization is applied. In particular, I fail to see any evidence that

The problem of generalization in neural networks:

My other concern is that the present results fail to contribute to a broader understanding of neural network generalization. Indeed, as the authors mention in the conclusion, strong regularizers are rarely used in practice, and do not seem to particularly affect the generalization properties of neural networks [see Zhang et al. 2017]. Instead, recent efforts in the community have attempted to explain the generalization of neural networks by identifying some distinguished and favorable property of the trained network: see e.g. [Arora et al. 2018, figure 2]. This can then be observed empirically on large networks, and a basic counterfactual can be established by measuring them on networks trained on random data or at initialization. On the other hand, while it is clear that bounding the capacity of the estimator (e.g. through a norm constraint in this case) yields favorable generalization properties, it is not clear to me how relevant these are to larger networks.

Minor comments:
1. Please include references for definitions (e.g. path-norm), and theorem headers when appropriate
2. The condition in Assumption 2 is often referred to as a “sub-gaussian tail / sub-gaussian noise”, rather than “exponentially decaying tail” (as in this particular instance, the tail decays as the exponential of a square).
3. After Lemma 2, in the derivation for L(\theta), the second equality sign should be an inequality. The last \hat{L}_{B_n}(\theta) should simply be L_{B_n}(\theta).
4. In table, the display is somewhat confusing: for accuracy, higher is better, whereas for \norm{\theta}_p / \sqrt{n}, lower is better. Consider expressing the accuracy as error instead.

---

### Meta-Review · Area_Chair1 · 2018-12-16
**Interesting contribution, but not quite ready**

**Confidence:** 4
**Recommendation:** Reject

**Metareview:**

I enjoyed reading the paper myself and agree with some of the criticisms raised by the reviewers, but not all of them. In particular, I don't think it's a major issues that this work studies an explicit regularization scheme BECAUSE the state of our understanding of generalization in deep learning is so embarrassingly poor!!

Unlike a lot of work, this work is engaging with the *approximation* error and developing risk bounds (called "generalization error" here ... not my favorite term for the risk!) rather than just controlling the generalization gap. The simple proof in the bounded noiseless case was nice to see.  On the other hand, not being familiar with the work of Klusowski and Barron (2016), I'm not willing to overrule the reviewers on judgments that this work is not novel enough. I would suggest the authors take control of this and paint a more detailed picture of how these two bodies of work relate, including how the proof techniques and arguments overlap.

Some other comments:

1. your theorem requires \lambda > 4, but then you're using \lambda = 0.1. this seems problematic to me.

2. your "nonvacuous upper bound" is path-norm/sqrt(n) ... but do the numbers in the table include the constants? looking at the constants that are likely to show up, (4Bn sqrt(2 log 2d), they are easily contributing a factor greater than 10 which would make these bounds vacuous as well.  you need to explain how you are calculating these numbers more carefully.

3. several times Arora et al and Neyshabur et al are cited when reference is being made to numerical experiments to show that existing bounds are vacuously large. But Dziugaite and Roy, who you cite for the term "nonvacuous", made an earlier analysis of path-norm bounds in their appendix and point out that they are vacuous.

4. the paper does not really engage with the fact that you are unlikely to be exactly minimizing the functional J. any hope of bridging this gap?

5. the experiments in general are a bit too vaguely described. also, you control squared error but then only report classification error. would be interested to see both.